# TPE-RBF-SVM Model for Soybean Categories Recognition in Selected Hyperspectral Bands Based on Extreme Gradient Boosting Feature Importance Values

Qinghe Zhao [ID], Zifang Zhang, Yuchen Huang and Junlong Fang *

Electrical Engineering and Information College, Northeast Agricultural University, Harbin 150030, China
* Correspondence: jlfang@neau.edu.cn

**Abstract:** Soybeans with insignificant differences in appearance have large differences in their internal physical and chemical components; therefore, follow-up storage, transportation and processing require targeted differential treatment. A fast and effective machine learning method based on hyperspectral data of soybeans for pattern recognition of categories is designed as a non-destructive testing method in this paper. A hyperspectral-image dataset with 2299 soybean seeds in four categories is collected. Ten features are selected using an extreme gradient boosting algorithm from 203 hyperspectral bands in a range of 400 to 1000 nm; a Gaussian radial basis kernel function support vector machine with optimization by the tree-structured Parzen estimator algorithm is built as the TPE-RBF-SVM model for pattern recognition of soybean categories. The metrics of TPE-RBF-SVM are significantly improved compared with other machine learning algorithms. The accuracy is 0.9165 in the independent test dataset, which is 9.786% higher for the vanilla RBF-SVM model and 10.02% higher than the extreme gradient boosting model.

**Keywords:** hyperspectral technology; non-destructive testing; soybean; machine learning; support vector machine; extreme gradient boosting; tree-structured Parzen estimator

## 1. Introduction

Soybean (scientific name: Glycine max), is an East Asian native in the legume family, whose seeds are an excellent source of plant protein and lipids [1]. Soybean has the dual properties of a food crop and an economic crop. About 85% of the global soybean crop is processed into soybean oil or soybean meal and the rest is processed in other ways or eaten directly [1,2]. Soybeans with insignificant differences in appearance have large differences in their internal physical and chemical components, so follow-up storage, transportation and processing require targeted differential treatment [3]. Therefore, there is an urgent need for fast automagical non-destructive testing technology to classify soybean varieties in the breeding, sorting and subsequent processing.

Hyperspectral technology, a non-destructive testing technology, is widely used in agriculture, the food industry and other many fields. Spectral data, compared with visual images based on traditional machine vision, can provide richer information from data sources. However, it is necessary for high-throughput spectral data to cooperate with effective analysis methods or advanced models to make the most of their rich amount of information in pattern recognition or outlier detection. Partial least squares regression and the PLS-DA model are traditional analysis methods for modeling and prediction: E.M. Abdel-Rahman et al. predicted chard yield grown under different irrigation water sources from hyperspectral data by improved sparse PLS regressions [4]; A. Folch-Fortuny et al. detected decay lesions in citrus fruits by N-way PLSR discriminant analysis model in SWIR bands of spectrum [5]; T. Rapaport et al. assessed grapevine water status by fusing both the hyperspectral imaging and leaf physiology with the PLSR model [6]. With the development of new machine learning technologies and the decline in computing power costs in recent years, the intersection

of hyperspectral technology and machine learning is getting more popular: L.P. Osco et al. proposed that random forest method performed well on predicted main elements and trace elements by hyperspectral data of Valencian orange leaves [7]; C. Erkinbaev et al. applied a perceptron model with backpropagation in artificial neural networks in the SWIR bands of wheat that tended to perform better than PLSR models in the hardness testing task of seed single grain [8]; Zhang X. et al. applied GA-SVM as the main algorithm with selected features in hyperspectral data using continuous projection of the NIR bands to complete adulteration identification for saline holothurian with explanations through spectroscopic analysis [9].

Support vector machine, a classic mature machine algorithm widely used in supervised learning tasks in various fields, has the advantages of both high robustness and great performance in most usages [10–12]. However, there is huge computing resource required when fitted and predicted by a high-dimensional dataset, such as hyperspectral data, so that a reasonable dimensionality reduction method is required as a pipeline before the data input. Direct dimensionality reduction based on mathematical calculation and sequential feature selection based on modelling are two more commonly used methods. LI Y. et al. applicated the principal component analysis in frequency domain spectral data for the SVM model to distinguish heavy-metal pollution in crops [13]; M. Pal et al. compared and discussed several feature selection algorithms, such as recursive feature elimination and correlation-based feature selection, for SVM methods in AVIRIS, a remote sensor in the hyperspectral dataset [14]; in addition, the hyperparameter configuration in the modelling of SVM is a key for better performance as well and it is necessary to achieve a limited number of model self-optimized iterations in an effective way. It is not only to improve the performance of the machine learning model itself, but also a crux link to realize the auto machine learning in a specific actual production environment in the future. In recent years, academic research, meta-heuristic algorithms, such as particle swarm optimization [15,16], simulated annealing algorithm [17], and genetic algorithm [16], has been more concerned with model optimization tricks, but random search or grid search is still widely used in practical deployed application [18,19]. Sequential-model-based global optimization is an optimization method that has been applied to large-scale neural networks, yet with better performance than traditional methods in practical engineering [19,20]. Further, it has been verified by this paper that it can also effectively improve performance in an SVM model with the hyperspectral sub-band dataset.

This paper proposes a multi-classification method for soybean seed by hyperspectral data based on support vector machine with Gaussian radial basis function kernel (RBF-SVM) optimized by a tree-structured Parzen estimator (TPE) after feature selection as follows:

- Dataset construction: the hyperspectral images range from 400 nm to 1000 nm, collected from 2299 soybean seeds and four categories were established;
- Feature selection using a boosting algorithm: an extreme gradient boosting algorithm was introduced to reduce redundancy dimensionality in the hyperspectral data. Ten feature bands were determined from the original 203 hyperspectral bands for a subset;
- Optimized RBF-SVM model with TPE: a support vector machine with Gaussian radial basis kernel function is built for the multi-classification pattern recognition task of soybean datasets and the tree-structured Parzen estimator method was introduced to improve the model as TPE-RBF-SVM.

Each category of soybean seed is able to differ by hyperspectral bands. The four categories multi-classification accuracy of the above TPE-RBF-SVM in the independent test dataset is 0.9165 (F1 = 0.9052), which is 9.786% higher for the vanilla RBF-SVM model without TPE and 10.02% higher than the extreme gradient boosting model. Compared with other machine learning algorithms, the metrics are significantly improved as well.

## 2. Materials and Methods

### 2.1. Soybean Material and Hyperspectral Dataset

The soybean samples were from the Soybean Research Institute of Northeast Agricultural University (Harbin, China). The basic nutritional information of four categories of soybeans is shown in Table 1. About 1000 seeds in each category were randomly selected primarily. After removing the samples with obvious abnormal appearance, the hyperspectral information of the samples was taken by Headwall VNIR-A system, as in Figure 1, in Application Research Laboratory of Spectral Technology of Northeast Agricultural University. In order to avoid jamming from the harmonic current generated by the large AC motor in the power supply system and further to limit the light source to the halogen light source of the experimental platform as much as possible, the spectral data were collected from 23:00 to 2:00 at night.

**Table 1.** Basic soybean information of nutrition and experiment dataset construction.

| Category | Crude Protein | Crude Fat | Shape | Seed Coat Luster | Seed Hilum | Train & Valid | Test Dataset | Label | Sum |
|---|---|---|---|---|---|---|---|---|---|
| Dongsheng-1 | 41.30% | 19.97% | Spherical | shiny | yellow | 375 | 125 | 0 | 500 |
| Changnong-33 | 37.57% | 23.00% | Ellipsoid | shiny | yellow | 374 | 125 | 1 | 499 |
| Changnong-38 | 37.26% | 21.33% | Ellipsoid | slight | yellow | 450 | 150 | 2 | 600 |
| Changnong-39 | 40.91% | 20.15% | Spherical | dull | brown | 525 | 175 | 3 | 700 |

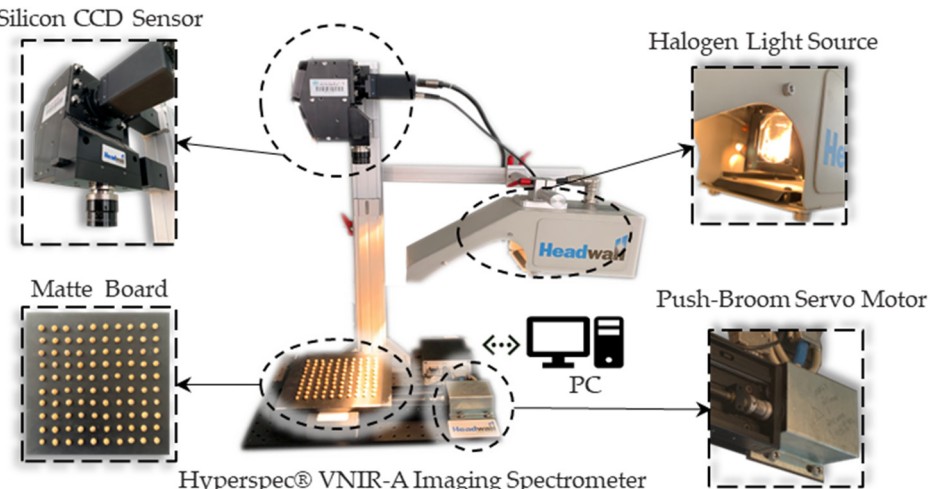

**Figure 1.** Headwall Photonics Hyperspec® VNIR-A system and customized matte board.

The shape of soybeans is spherical or ellipsoid so a push-broom board matching the shape of the seeds was designed to maintain the stability before taking images. The customized board is a square with a side length of 200 mm and a thickness of 10 mm made by polypropylene. The surface of the board is subjected to diameter of 10mm and depth of 10mm drilling to ensure that the soybeans can be fixed in the holes. In order to avoid scattering of the halogen light source by the polypropylene material, the surface was further sprayed with water-based acrylic paint (Botny-B1924-#4) for matte treatment and then finished with a dark-white correction in blank board.

The sample collection wavelength is configured from 400 nm to 1000 nm, the ultraviolet-visible-near-infrared spectral band, the width of bands is about 3 nm and, in total, 203 bands are collected. During acquisition, the moving speed of the push-broom board is 5 mm/s with 38.84 ms exposure time. The imaging range is controlled within the extinction plate and the frame period is 0.04 ms. The final hyperspectral cubic is obtained with a size of 1004, 812 and 203.

The soybean seed information was exported from the cube and transform to data with instrument response for further processing by ENVI Classical 5.3. The ROI area is defined by masked method and the average sample hyperspectral data are obtained by taking a



single soybean seed as an independent sample. Figure 2 shows the average spectral curve of the four categories of soybean. The vertical axis is the instrument response value that is proportional to the reflectance of the spectral data. Because data preprocessing and format conversion are involved in the later stage, the instrument response is directly applicated as raw data. Finally, the dataset is split into training, validation and test dataset according to a certain ratio (2:1:1) as shown in Table 1.

**Figure 2.** Mean spectral curves for four soybean categories.

*2.2. Support Vector Machine Model and Optimization*

2.2.1. Support Vector Machine with Gaussian Radial Basis Kernel

Support vector machine (SVM) is a robust supervised learning algorithm based on statistical machine learning theory jointly proposed by V. N. Vapnik and A. Y. Chervonenkis [21]. The SVM algorithm iteratively searches the hyperplane in the sample space of data and obtains a lossy partitioned hyperplane interval composed of support vector (SV) to complete the classification task in machine learning; B. E. Boser and V. N. Vapnik thereafter introduced the kernel function into the SVM to map the original sample space in the high-dimensional Hilbert space for further optimization of the model application [22]; Chih-Jen Lin et al. programed libsvm and liblinear, an efficient implementation of the quadratic programming (QP) in SVM and then algorithm was widely used in pattern recognition and other fields in machine learning [23].

From the perspective of statistical learning, when the sample is like $\{(x_i[m], y_i), i = 1, 2, 3, \ldots, n\}$, composed of $n$ samples of $m$ dimension and corresponding real values $y$, the support vector machine needs to solve the following, Equation (1), iteratively:

$$\underset{\omega, b, loss}{argmin} : \frac{1}{2}\omega^T\omega + C\sum_{i=1}^{n} loss_i \text{ ; s.t. } y_i\left(\omega^T\psi(x_i)+b\right) \geq 1 - loss_i, \; loss_i \geq 0 \qquad (1)$$

where $\omega$ and $b$ are the parameter of point-normal equation of a hyperplane, $y = \omega x + b$; $loss_i = max\left(0, 1 - y_i(\omega^T\psi(x_i) - b)\right)$ is the hinge loss of true values and predicted ones in sample space; $C$ is the strength of the regularization of objective function; $\psi(x_i)$ is the map function between sample space of $x_i$ and high-dimensional Hilbert space that introduces kernel function method later.

The objective function (1) is a convex optimization problem, which satisfies the Karush–Kuhn–Tucker conditions, that is commonly solved by the Lagrangian dual method [21,24]. The objective function after conversion of the parameters to the dual problem is as follows (2):

$$\underset{\omega, b, loss}{argmin} : \frac{1}{2}\alpha^T Q\alpha - e^T\alpha \text{ ; s.t. } y^T\alpha = 0, \; 0 \leq \alpha_i \leq C \qquad (2)$$

where $Q$ is the transformed positive semi-definite Hermitian Matrix, whose elements are $Q_{ij} = y_i y_j \psi(x_i)^T \psi(x_j)$; $e$ is the one vector full with the number 1 to ensure the validity of the calculation; $\alpha$ is the dual coefficient vector.

The dual objective function (2) uses Quadratic Programming to complete the iterative solution of $\alpha$ and the support vectors $SV = \{x_k, k \in QP(\alpha)^*\}$. In the solution and application process, the Gaussian radial basis kernel function (RBF) is introduced for mapping the samples into the Hilbert space. For the prediction $\hat{y}$ of a new sample $(x, y)$, it can be expressed in the following form (3):

$$\hat{y} = sgn \sum_{i \in SV} y_i \alpha_i \kappa(x_i, x) + b \tag{3}$$

where $\kappa(x_i, x) = \psi(x_i)^T \psi(x) = exp(-\frac{||x_i - x||}{2\sigma^2})$ and the σ is a free parameter to control the mapping process in Gaussian radius basis function [24].

That is, the general or vanilla Gaussian radial basis kernel function support vector machine, short for RBF-SVM.

### 2.2.2. Optimization of SVM with Tree-Structured Parzen Estimator

Based on the derivation (1) to (3) of the above mathematical principles, the proven SVM model needs to artificially determine the penalty scaling strength C in Equation (1) and the free parameter σ of the kernel function control mapping in Equation (3). It is customary to express σ with $\gamma = \frac{1}{2\sigma^2}$ and γ is usually set as $\frac{1}{m}$ or $\frac{1}{m} \times \sum \frac{(x_i - \bar{x})^2}{n}$. according to the empirical formula for datasets with $n$ samples and $m$ features [25]. Penalty scaling strength C generally is determined as $10^k$ ($k \in N$) experimentally in the validation dataset or cross-validation [25].

Optimization for hyperparameters can be further determined to verify the prediction performance of the SVM as follows (4):

$$\underset{C,\gamma}{argmin} : L(y_i, \hat{y} = f(x_i, C, \gamma)), (x_i, y_i) \in \textbf{valid} \tag{4}$$

where $L$ is a metric (accuracy in this paper) or a loss function to evaluate model performance in validation dataset; $f$ is the fitted RBF-SVM model with corresponding hyperparameters, C and γ, in optimization.

Objective function (4) can be seen as an optimization issue of best hyperparameters pair $\mathbf{z} = (C, \gamma)$ in the sample space in essence of black-box process [19]. We will implement this by a Bayesian optimization method, the tree-structured Parzen estimator (TPE). It is a sequential-model-based optimization (SMBO) method proposed by Ozaki et al. in 2020 that originally optimizes for large-scale neural networks [20]. The pseudo code of Tree-structured Parzen Estimator method in Algorithm 1 is as follows:

---
**Algorithm 1** The pseudo-code of tree-structured Parzen estimator algorithms (for RBF-SVM)

---
1: Initialization $\mathbf{H_0} = \emptyset$, $\mathbf{z} = \mathbf{z_0}$
2: For: k = 1 to $I_{max}$
3: Update hyperparameters: $\mathbf{z}^* = argmin(EI_k(P, \mathbf{z}_k[H_{k-1}]))$
4: RBF-SVM repeat fitting and evaluating: $L_k$
5: Update optimization history: $\mathbf{H}_k = \mathbf{H}_{k-1} \cup <EI_k, L_k>$
6: End
7: Return $\mathbf{z}* = argminL(y_i, \hat{y} = f(xi, C, \gamma))$

---

TPE algorithms build a probabilistic model for optimization objective function targeting $\mathbf{z}$ by the surrogate function, $EI$ values and threshold $P$ in iteration history and evaluate the results of tuning values $L$ in validation dataset or cross-validation method. The surrogate function is as follows [20]:

$$EI = \frac{\int_{-\infty}^{+\infty} \max(L * -L[H_{i-1}], 0) \times p(L[H_{i-1}])dL}{P + (1 - P) \times \frac{g(z[H_{i-1}])}{h(z[H_{i-1}])}} \tag{5}$$

where $g(z)$ and $h(z)$ are the probability distributions when the value of $L$ is greater than or less than the threshold $P$, respectively, whose distributions come from the historical information $H$ accumulated in the previous $k - 1$ iterations.

The combination of TPE method and RBF-SVM algorithm realizes hyperparameter tuning and model optimization based on SMBO of Bayesian optimization according to the above theory. When the training, validation and test dataset meet the independent and identically distributed (IID), this method will balance the empirical risk and generalization risk of the model to the greatest extent and finally complete a more accurate modeling process through the automatic configuration of controlled hyperparameters.

*2.3. Crux Spectrum Feature Selection Based on Extreme Gradient Boosting*

The spectral data are high-throughput sequencing that would cause dimensional explosion when support vector machine directly applicates the kernel function to map raw data into a higher-dimensional space that heavily depends on computing power and time. In addition, there is high redundancy for hyperspectral information itself and not all bands have a positive contribution to fit the model in classification task. Key feature selection of raw data is necessary to extract more crux information than RGB image and less redundancy for model building for precision fitting; therefore, a method based on boosting algorithm is adopted in this paper.

Gradient Boosting, a boosting branch strategy of ensemble learning methods, was originally proposed by J.H. Friedman, who combined series of weak learners called meta learners with the specific strategy for single machine learning task [26]. The final model will have better performance than single meta learners ensembled. Extreme Gradient Boosting is a gradient boosting algorithm proposed by Tianqi Chen in 2011 [27]. The algorithm is constructed by CART trees as the main meta learner and set the quadratic Taylor expansion of the loss function in the boosting iteration. Model based on trees has the better interpretability and ensemble process more effectively improves the performance than single tree model so that it has been widely applicated and deployed in various fields requiring reliance on feature interpretability.

As an additive model, objective function of the meta learner $f$ of the single round in the dataset $(x_i, y)$ is as follows:

$$obj(t) \ = \sum_{i=1}^{n} \boldsymbol{loss} \left[ y_i, \hat{y}_i^{t-1} + f_t(\boldsymbol{x}_i) \right] + \sum_{i=1}^{T} \Omega(f_i). \tag{6}$$

where $t$ is the boosting rounds in model and $\Omega$ is the regularization part of the tree.

After the quadratic Taylor expansion of the loss function and the structural parameters of the CART tree occur, the objective function is as follows (7):

$$obj(t) = \sum_{i=1}^{n} \left[ g_i^t \cdot f_t(\boldsymbol{x_i}) + \frac{1}{2} \cdot h_i^t \cdot f_t(\boldsymbol{x_i})^2 \right] + \gamma \cdot T + \frac{1}{2} \cdot \lambda \cdot \sum_{j=1}^{T} \omega_j^2 \tag{7}$$

where $g$ and $h$ are first and second derivatives of loss function, $T$ is the number of the leaves of tree, $\omega$ is the weight of tree leaf and $\gamma$ is the minimize gain for node to split. All three are the construction of CART meta model $f$, $\lambda$ is the L2 regularization from $\Omega$.

Reindex to the objective function (7) from rounds' $t$ to nodes' $j$ is solved by the greedy strategy to get best solution $\omega_j^{[best]} = -\frac{G_j}{\lambda + H_j}$ where G and H are the sum of the derivatives in the loss functions and return to (7) after simplification as follows (8):

$$best_{obj(t)} = -\frac{1}{2} \cdot \sum_{j=1}^{T} \frac{G_j^2}{H_j + \lambda} + \gamma \cdot T \tag{8}$$

The loss or the metrics in the training dataset will continue improving as the number of rounds increases if the ensemble model is trained as in (8). However, too many rounds would lead to overfitting. Early stopping can control the over training of the

boosting that would stop the boosting iterating when cross-validation or validation dataset cannot improve after the rounds we set and output the best numbers of iteration as the final ensemble.

For each meta CART tree learner in boosting, the gain in feature splitting is calculated as follows:

$$\text{Gain} = \frac{1}{2}\left[\frac{G_L^2}{H_L+\lambda} + \frac{G_R^2}{H_R+\lambda} - \frac{(G_L+G_R)^2}{H_L+H_R+\lambda}\right] - \gamma \tag{9}$$

where L and R mean the splitting path and the calculation method are the same as above (7) and (8) and Gain is the splitting indicators or references to grow a tree with dependence maximization of features in sample space. The ensemble model makes a mean calculation of all $t*$ estimators for macro-Gain as **FIV** to measure feature importance in the model [28]. The FIVs are as follows (10):

$$\text{FIV}(s) = \frac{\sum_{i=1}^{t*}(\text{Gain}_s)}{\sum_{i=1}^{t*}\sum_{j=1}^{m}(\text{Gain}_j)} \tag{10}$$

The value of **FIV**($s$) represents the information gain of the feature $s$ to all meta learners or models during the iteration of the extreme gradient boosting. When the performance of the model reaches a certain acceptable level, **FIV**s can be used in both feature selection and data interpretation. The range of this value is a floating number in (0.00, 1.00). The closer it is to the value of the right boundary, the more important the feature is for the ensemble model. This paper will use the FIV value model based on the extreme gradient boosting algorithm for feature selection, compress the high-dimension spectral data into fewer crux bands and then build RBF-SVM model with the TPE method mentioned above.

### 2.4. Feature Selection and Optimized RBF-SVM Modelling

The model design is as shown in Figure 3. The TPE-RBF-SVM model for the feature-band subset is designed in two steps:

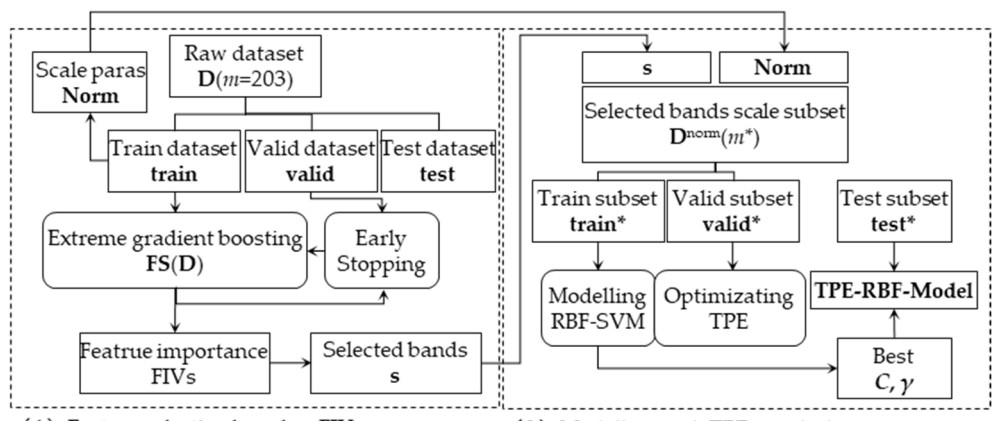

**Figure 3.** Flow chart of feature selection and optimized RBF-SVM modelling.

### 2.4.1. Feature Selection Based on Feature Importance

Feature selection transforms the 203-band UV-Vis-NIR spectral dataset **D**($m$ = 203) into a wavelength band subset **s** with 10 crux spectral bands.

Firstly, the extreme boosting model is built and fitted by full-band dataset train of **D**($m$). The fitting process would have a validation in **valid** dataset of **D**($m$) to control possible overfitting and underfitting, then a full-band extreme boosting model **FS**$_{\mathbf{D}(m=203)}$ was finished. After performance evaluation, the **FIV**s of acceptable model, **FS**$_{\mathbf{D}(m=203)}$, will be extracted and mask for a subset by the sorted descending top 10 spectral band wavelengths as **s** with feature interpretation.

2.4.2. Modelling and Optimizing RBF-SVM with TPE in Sub-Dataset

According to the characteristics of the SVM algorithm, the original data **D**(*m*=203) will be scaled as follows:

$$x_i^{norm}(j) = \frac{x_i(j) - \overline{x(j)}}{x_{\text{std}}(j)}, i = 1, 2, 3, \ldots, n; \; i = 1, 2, 3, \ldots, m \tag{11}$$

where $\overline{x}$ and $x_{\text{std}}$ are the average and variance value of the dataset train correspondingly and these data will be used directly as constant values in normalized scaling for **valid** and **test** to avoid data leaks that pollute the training dataset by accident information.

After the scaling, **D**$^{norm}$(*m\**=10) is constructed according to sub-dataset in (11) above. The RBF-SVM is going to build and be fitted by **train\*** of **D**$^{norm}$(*m\**=10) with optimization searching by TPE algorithms in **valid\*** datasets for better performance metrics.

The RBF-SVM algorithm here is built by the libsvm with OVR (one vs. rest samples) for multi-classification. That is transformed into 4 two-class sub-tasks and each two-class task recognizes the single category with all other categories. In order to avoid overfitting from extreme or outlier samples in the modelling, the maximum number of searching support vectors is constrained to 5000 times.

Table 2 shows the search space of the TPE-RBF-SVM including the hyperparameter C (penalty scaling strength of SVM) and γ (kernel coefficient of RBF) as follows:

**Table 2.** Hyperparameters to tune and search space.

| Hyperparameter | Data Type | Search Space | Minimize Step |
|----------------|-----------|--------------|---------------|
| C | float | $1 \times 10^{-2}, 1 \times 10^{5}$ | $1 \times 10^{-8}$ |
| γ | float | $1 \times 10^{-8}, 1$ | $1 \times 10^{-10}$ |

*2.5. Baseline Models and Evaluation Metrics Design*

2.5.1. Vanilla and Control Group Models

The extreme gradient boosting model itself is a supervised learning algorithm, which will be further tested as a comparison algorithm (xgbc) in the subset **D**(*m\** = 10). Further, Vanilla RBF-SVM (svc2) will be considered to compare the effect of the TPE; in addition, six other machine learning algorithms are shown as comparative models below:

- CART Tree (tree)

Decision tree is a non-parametric supervised learning model commonly applicated in machine learning. CART, one of decision trees, is also a meta learner in the most boosting model. In this research, the Gini index is set as the information gain indices for the comparison model, whose maximized depth of the tree is not constrained, node splitting method is the greedy and all features are considered during the splitting nodes.

- Random Forest (rdrf)

Random forest is another popular ensemble model based on tree model in engineering applications. By controlling the hyperparameters of the meta learners, the model randomly splits the feature of the samples and uses random sampling for the sub-sample to expand the dataset. To control model's variables and hyperparameters were configured as the same with boosting model, but did not limit the max depth of the tree.

- Logistic Regression (lgst)

Logistic regression is a probabilistic classification method that maps a function of dataset features to a target to predict that a new example belongs to one of the target categories' models learnt. It is a classic linear classifier in machine learning. In the model we built, the L2 regular term is added to the iterative objective and the maximized number of iterations for convergence is limited to 100. The solution method is configured as LBFGS. Logistic regression model is sensitive to the size of the pre-experimented data, so the standardized D$^{norm}$(*m\** = 10) is used for modeling.

- Multilayer Perceptron (mlp2&mlp4)

Perceptron is a widely used model in deep neural network, which adopts back-propagation to realize iterated learning by given data features. Perceptron has excellent learning ability for nonlinear tasks or datasets. In this paper, two kinds of perceptron with 2 and 4 layers compiled by Adam are used as a comparison of neural network. The activation function of the hidden layer is configured as RELU function, the number of hidden nodes is (100, 50) and (128, 64, 32, 16), the output layer is with SoftMax function and the learning rate is configured as 0.001. We limited the maximum iterations to 200.

- Convolution Neural Network (conv)

Convolution Neural Network is a powerful back-propagation deep learning algorithm and some research [29,30] about soybean with NIR bands (over 800 nm) show the potentiality in general spectral data analysis. Thus, we think it is necessary to test the network in our 400 to 1000nm bands. Considering our dataset's character we designed a multi-layer 1D-CNN model as the baseline. The CNN model compiled by Adam optimizer has two 1D-conv layers with 100/50 filters and 4/4 window length, then the dropout layer (rate = 0.1) is connected after each one to avoid overfitting. The final output layer is a Perceptron with SoftMax function after a flatten. The iteration loss is categorical cross entropy so we transform the probabilistic output to label-coded one before evaluation.

### 2.5.2. Evaluation Metrics and Analysis Environment

The identification task among soybean categories is a supervised machine learning multi-class task; therefore, the multi-class accuracy is used as the main evaluation metrics firstly for the model in this paper. The multi-class accuracy ACC is calculated as below:

$$\text{ACC} = \frac{1}{n_{\text{sample}}} \sum_{i=1}^{n_{\text{sample}}} I(\hat{y}_{\text{i}} = y_i) \tag{12}$$

where $\hat{y}_{\text{i}}$ is the predicted value of corresponding true value $y_i$ and the $n_{\text{sample}}$ is the number of samples in dataset. Further, an ACC-based confusion matrix will be introduced for detailed category prediction evaluation.

On the other hand, the research will additionally use F1 score as the second evaluation metric. The F1 score is the harmonic mean of the precision and recall from the dataset. The calculation method is as follows:

$$F_\beta = \left(1 + \beta^2\right) \times \frac{precision \times recall}{\beta^2 \times precision + recall}, \left(precision = \frac{TP}{TP + FP}, \ recall = \frac{TP}{TP + FN}\right) \tag{13}$$

where $\beta = 1.00$ means the harmonic mean calculation, the *TP* is the counts of correctly predicting as True Positive and the *FN*, False Negative, is the counts of misrecognition.

All the codes designed in this research were open sourced under the MIT license. The hardware environment that the analysis depends on is shown in Table 3 below and the compile environment is based on Python 3.9 in Windows 10 LTSC. To ensure reproducibility in all the experimental and analysis results, all random seeds involved in this paper are set as 615.

**Table 3.** Environment and tools of analysis and model building in paper.

| | Compute Environment | Analysis Tools |
|---|---|---|
| CPU | Intel® Core™ i5-10400F (2.90 GHz) | Pandas 1.3.3, Numpy 1.19.3, |
| GPU | Nvidia GeForce RTX 3070 | Scikit-learn 0.24.1, |
| RAM | DDR4 3000Mhz 48GB = 2 × 8 GB + 2 × 16GB | XGBoost 1.4.2, Scipy 1.6.2, |
| Operating System | Windows LTSC 21H2 | liblinear 3.23.0.4, |
| Random Seed | 615 | CUDA 11.2, Keras 2.9.0 |

## 3. Results

### 3.1. Feature Selection Based on FIVs

The training dataset, **train**, in a range of 400 nm to 1000 nm, fits the extreme gradient boosting model cited above, which applicated **valid** in a separate validation dataset for early stopping. The boosting was set to the max number of estimators of 1000 rounds and finally the iterations stopped in the 880th round. At this time, the model with the validation dataset would not have been improved by the loss metrics in 10 rounds so it is deemed to fit fully. $\textbf{FS}_{\textbf{D}(m=203)}$ and corresponding **FIV** can be pick up from the fitted boosting model.

Figure 4 shows the distribution of bands and **FIV**, whose horizontal axis represents the band wavelength of the spectral data; the vertical axis is the **FIV** numerical distribution with a sum of 1.00 from the extreme gradient boosting model. According to the **FIV** value, the top 10 wavelengths are extracted as the selected crux wavelength band to **s**, whose distribution is shown covered in the band marked with the dotted line.

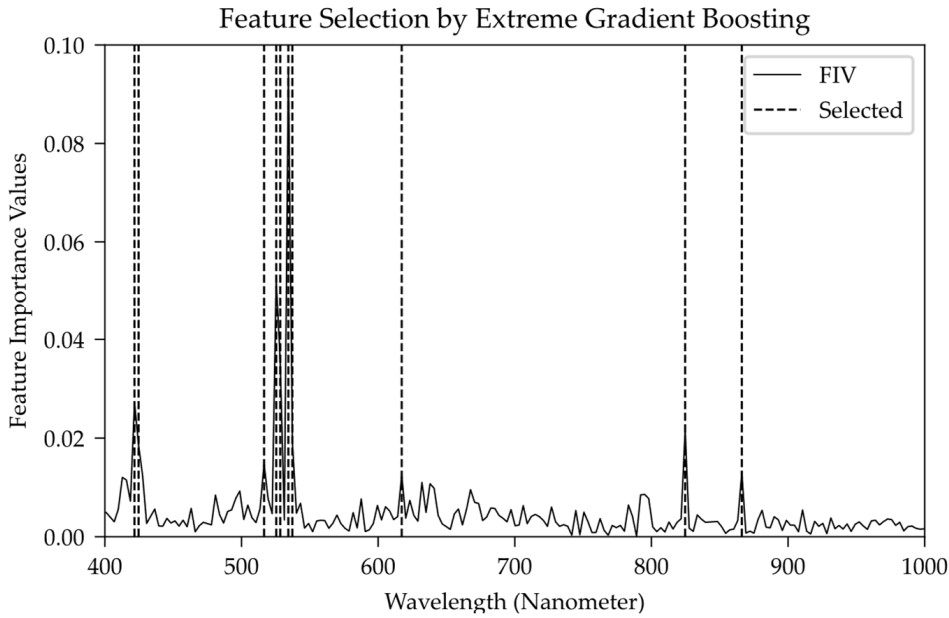

**Figure 4.** All bands' FIV values and the selected ten crux bands.

Table 4 shows the central wavelengths of 10 selected bands and their **FIV**. Ten selected bands contribute 30.9631% in the model. The average weight of each band is calculated to transform it into relative FIVs based on the sum of selected bands' values.

**Table 4.** Selected bands of center wavelength and relative values.

| Center Wavelength | Violet Bands | | Green Bands | | | | Orange | NIR Bands | | |
|---|---|---|---|---|---|---|---|---|---|---|
| | 421.66 | 424.63 | 516.49 | 525.38 | 528.35 | 534.27 | 537.24 | 617.25 | 824.69 | 866.18 |
| Full Bands' FIV values | 2.73% | 1.86% | 1.48% | 5.23% | 3.52% | 9.51% | 1.92% | 1.25% | 2.16% | 1.31% |
| Relative FIV values | 8.80% | 6.01% | 4.77% | 16.88% | 11.38% | 30.71% | 6.19% | 4.04% | 6.98% | 4.22% |

Combining Table 4 and Figure 4, the selected crux bands include two violet bands (VLNs), five green bands (GRNs), one orange band (ORG) and three near-infrared bands (NIRs). Among them, the green bands (GRNs) account for 21.66% of the whole band (relative value is 69.93%), which is the dominant spectral band in the extreme gradient boosting model and there are several obvious peaks in Figure 4. The violet band (VLT) and the near-infrared band (NIR) accounted for 4.59% (14.81%) and 3.47% (11.20%), respectively, which can be considered as two spectral bands with the same secondary modeling importance.

The instrumental response subset bar plots of hyperspectral data from boosting model for four classes of soybean are shown in Figure 5, where the right side lists six visual pictures in 421.66 nm, 534.27 nm, 617.25 nm, NIR (824.69 nm), color infrared (CIR, 534.27 nm/824.69 nm/421.66 nm) and true color image (RGB). Through the image display of the first three bands, the hilum and other visual features of soybeans can be identified in the visible-light band. The NIR directly crosses the texture band and further shows the refraction and projection of the light source by the intensity of the instrumental response value. CIR and RGB are indirectly or directly visible to human vision. It is difficult to distinguish the four classes of soybeans, but with the help of selected bands, it is more straightforward to complete the judgment of soybean types through data quantification.

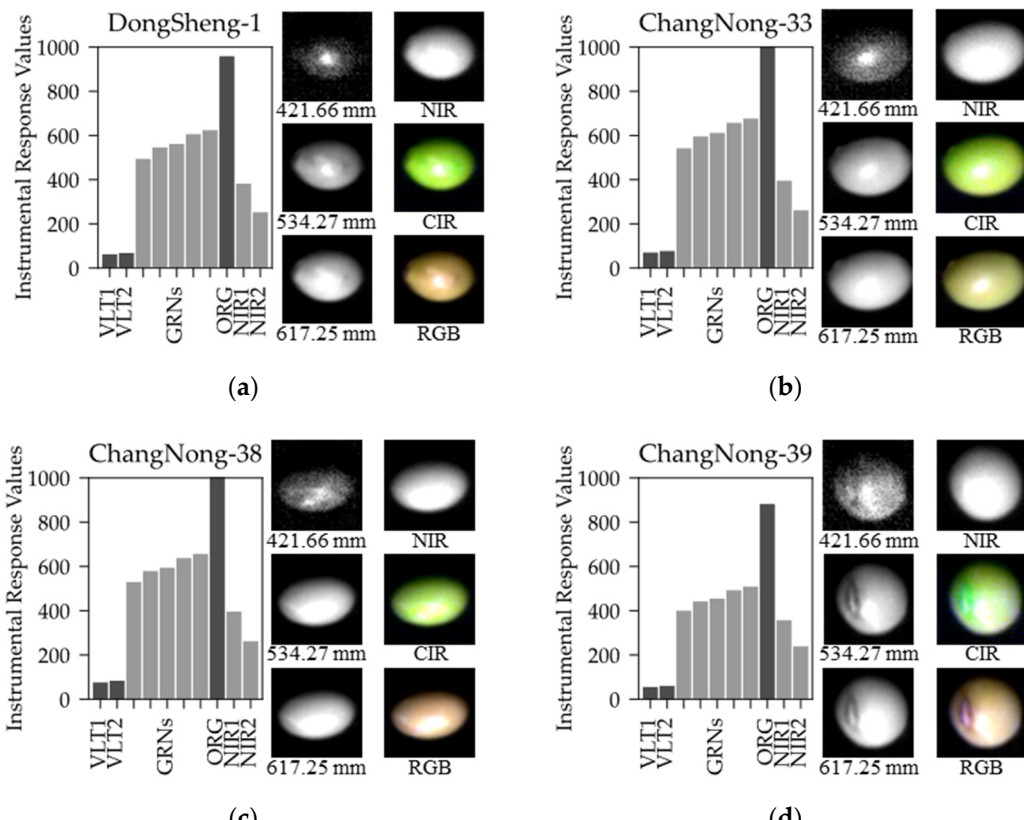

**Figure 5.** Subset and visual features image for soybeans in this paper. (**a**) DongSheng-1 subset (Label 0). (**b**) ChangNong-33 subset (Label 1). (**c**) ChangNong-38 subset (Label 2). (**d**) ChangNong-39 subset (Label 3).

### 3.2. Optimization of SVM by Tree-Structed Parzen Estimator

In the scaled feature subset train∗, we kept the same 72 iterative searches as grid search and the optimized hyperparameters in the final saved iteration results are $C = 1.7631 \times 10^4$ and $\gamma = 2.1056 \times 10^{-4}$, which appears during the 32nd iteration. At this time, the ACC in the validation dataset is 0.9072 and the ACC after testing in the independent dataset **test\*** is 0.9165. The two values are similar so the generalization ability of the TPE-RBF-SVM model with both structural risk and empirical risk is proved.

Figure 6 is the distribution of the ACC metric in the validation process as the number of tuning iterations increases. Obviously, the ACC has a large variation range in the first 20 iterations of the search process. It can be explained as TPE is accumulating the researching statistics in the defined space, as Chapter 2.2. cited, so the metric fluctuates wildly. In the subsequent 20 to 40 iterations, the fluctuation of the metric became slighter, but the overall effect was generally reduced, which showed that the ACC value diverged again. This is perhaps due to the over-search caused by further iteration after the statistical

results. Then, until the 72nd iteration, the metric value is stable after several rounds of shocking and the best result appears in 20~40 between the two mutations.

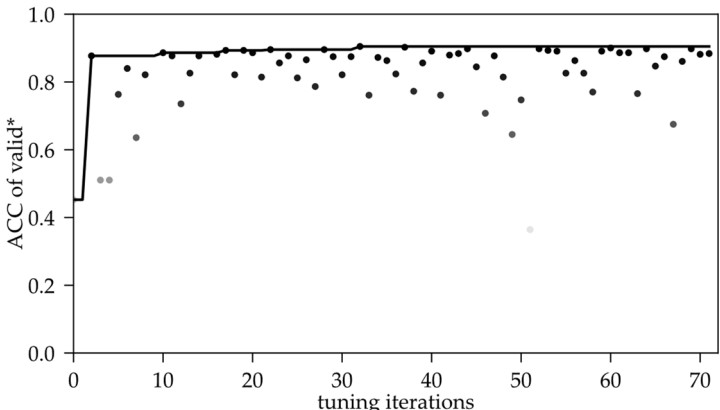

**Figure 6.** Accuracy and iterations of TPE optimization process.

Figure 7 shows the optimizing process tuned by C and $\gamma$; the color depth of the point increases with the number of iterations. Both hyperparameters have a tendency to gather towards a certain center, where the best combination (C = $1.7631 \times 10^4$, $\gamma = 2.1056 \times 10^{-4}$) is close to in the search space. According to the algorithm theory of SVM, when the training dataset is complex multi-dimensional data, the relationship is complex between parameters of SVM and hyperparameters during the training process. Therefore, results of tuning by the metric will be of multiple local optimal solutions in the search space. This is confirmed by several iterations in the convergence–mutation process of the metric from validation by TPE as the number of iterations increases.

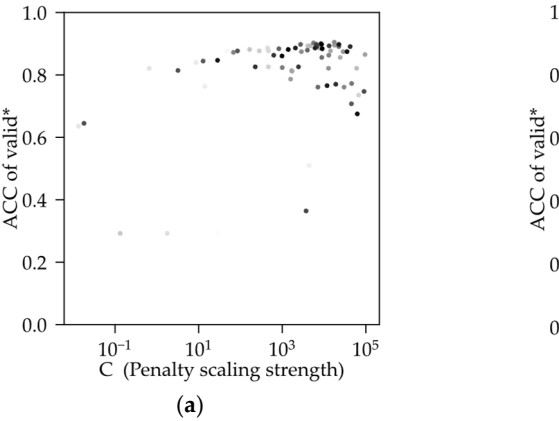 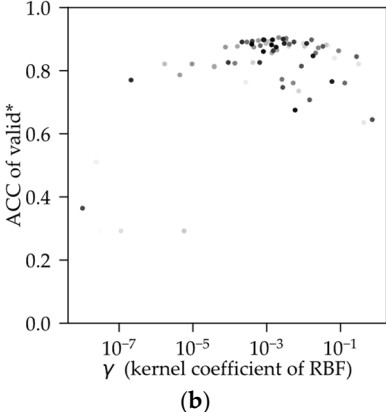

**Figure 7.** Hyperparameters changing during iterations. (**a**) Change for accuracy with C and iterations. (**b**) Change for accuracy with $\gamma$ and iterations.

The hyperparameters of the RBF-SVM model were finally determined by the TPE algorithm in the sub-band spectral dataset in 72 iterations. The model hyperparameter C = $1.7631 \times 10^4$ and the hyperparameter $\gamma$ is determined to be $2.1056 \times 10^{-4}$ for our datasets. The accuracy of the TPE-RBF-SVM model under this configuration is 0.9072 in the validation dataset and 0.9165 in the test dataset.

### 3.3. Comparison with Vanilla Model and Other Algorithms

The sub-band dataset, obtained in Chapter 5.1, is used as training data fit the SVM with the tuned optimized hyperparameters obtained in Chapter 5.2 to build the TPE-RBF-SVM model (bst, short of "best"). The same training data are used to fit the other eight models cited in Chapter 4.3. Then, the model is tested with the individual test dataset for metrics. ACC and F1 scores are shown in Figure 8.

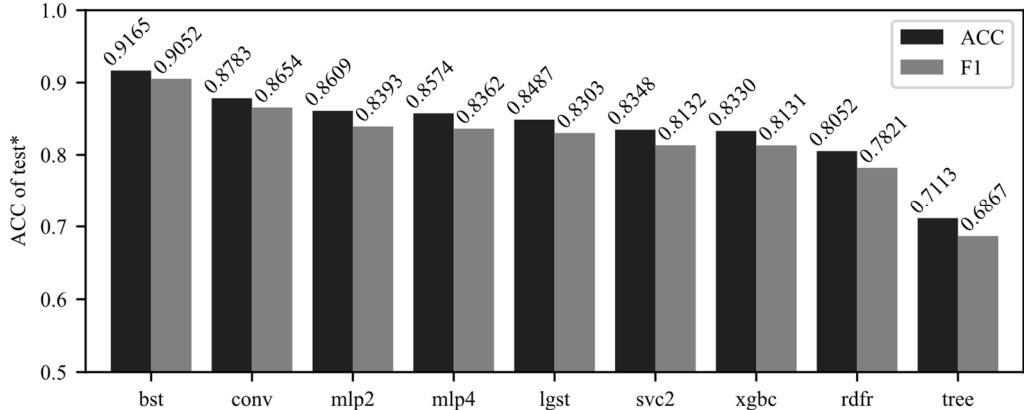

**Figure 8.** Accuracy and F1 score of all models in research.

The TPE-RBF-SVM model we recommended scores the highest in both metrics with four-classification accuracy of 0.9165 and F1 scores of 0.9052. Compared with unoptimized vanilla RBF-SVM, its accuracy improves by 0.0817 from 0.8348 (9.786% increasing in percentage). Compared with the second-ranked CNN model (conv), the accuracy is increased by 0.0382 (4.349% increasing) and the F1 score is increased by 0.0398 (4.599% increasing). For our hyperspectral dataset, both shallow and deeper network models perform similarly. In the feature subset data, there is a very obvious improvement effect. It can be preliminarily considered that the method we recommended has a very obvious improvement effect in selected sub-datasets for soybean pattern recognition.

Compared with the vanilla extreme gradient boosting model, the accuracy of the TPE-RBF-SVM model increased by 0.0835 from 0.8348 (10.02%). The results show that when the sub-band dataset is obtained by the boosting model and model built as a core classifier in this study, the accuracy is better than another ensemble model, random forest (accuracy = 0.8157) and meta learner, decision tree (accuracy = 0.7113). However, the boosting model does not significantly outperform the experimental results of other types of algorithms and is comparable to the unoptimized vanilla SVM model.

The confusion matrices of the TPE-RBF-SVM, the vanilla RBF-SVM and the extreme gradient boosting model are shown in Figure 9. The vertical axis of the confusion matrix is the real category and the horizontal axis is the predicted category from the model. The confusion matrices show that both the SVM models and the boosting model have excellent results for category 2 (Changnong-38) and category 3 (Changnong-39) and the TPE-RBF-SVM can almost recognize them totally. However, on the other hand, category 0 (Dongsheng-1) and category 1 (Changnong-33), which are more difficult to classify correctly, almost contribute all the negative effects of the three models, while the tree-structured Parzen estimator algorithm significantly suppressed this effect, with excellent results among the models.

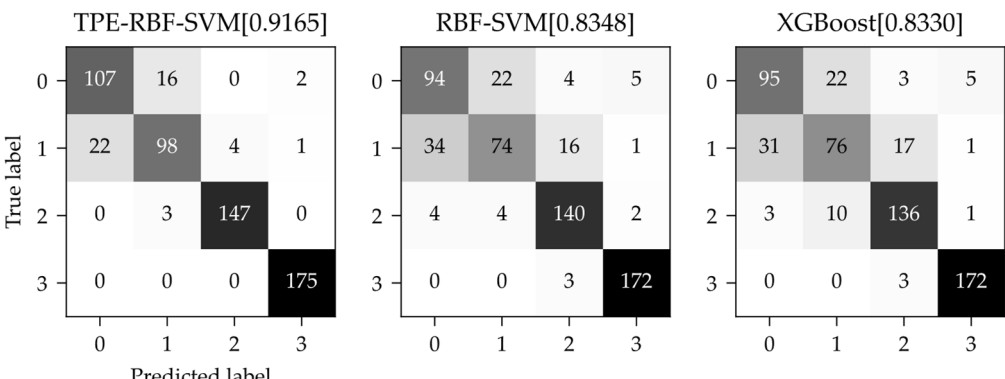

**Figure 9.** The confusion matrices of classification results of TPE-RBF-SVM and vanilla models.

**FIV**s based on the extreme gradient boosting method can select crux bands from bands ranging from 400nm to 1000nm for dimension-reduced sub-band datasets. Further, in the sub-band dataset by FIVs from extreme gradient boosting, compared with the vanilla RBF-SVM and XGBoost models, the SVM model optimized by TPE algorithms can effectively improve in the test dataset performance of soybean multi-classification tasks; compared with other machine learning algorithms, the method still has high accuracy as well.

## 4. Discussion

In this study, we demonstrated that optimized TPE-RBF-SVM is improved after feature selection by feature importance values. As shown in Figure 9, the model we recommend has better performance than either vanilla RBF-SVM or extreme gradient boosting. The feature selection plays a key role in this process firstly, because there are complex effects for the SVM algorithm when it is applied in datasets with highly redundant features [14] and the selection by boosting method directly controlled counts of dimension before fitting [31]. Thus, the negative effects from data collinearity and information redundancy can be avoided for SVM. Secondly, the extreme gradient boosting is a popular algorithm with wide applications, but not all boosting models or tree-based models are suitable for any application. After the corrected selection model in the first stage, all we need is a useful learner or machine learning model to classify or regress for the task. For the dataset in this paper, SVM indeed performed better than the boosting model after dimension reduction. The research paper of [32] discussed similar views, that in some applications with fewer features, SVM can perform better than Ensemble algorithms.

Furthermore, the combination between two algorithms is not casual but conditional and when the model for FIVs is over-fitted or under-fitted, the subsequent feature selection method expands the error because the selection model has been based on inappropriate feature weights. In the preliminary stage of data exploration by the boosting model in the paper, only the model itself has an acceptable performance and a certain generalization ability and its own FIV value is meaningful for feature selection. These phenomena, similar to the research papers in [28,31], can confirm each other.

Lastly, the final choice in our research is based on feature selection for dimensionality reduction and direct dimensionality reduction methods, such as principal component analysis, are not considered for research and comparison, because a method that can directly determine the spectral band is desired and there is no need to collect useless bands from a device-side source [33,34]. That is, when we applied the TPE-RBF-SVM method in a practical engineering issue, only several bands we considered, as in Figure 5 or Table 4, should be collected as a cheaper way in agricultural fields and bands we selected can be explained by Ultraviolet-visible-NIR Spectroscopy theory. For the same reason, this paper does not use conventional spectroscopy pre-processing, such as SNV (standard normal variate), MSC (multiplicative scatter correction) or Savitzky–Golay filter smoothing. All were cited before needing batch information for the whole dataset and, for single samples, we cannot do it because full-band data still need to be acquired. Further, there is a risk of data leakage when conducting dependent test datasets in research before the application stage.

## 5. Conclusions

With the results above, each category of soybean seed is able to differ by hyperspectral bands. A fast and effective machine learning method was designed based on less bands' hyperspectral data of soybean for pattern recognition of categories, designed as a non-destructive testing method. The conclusions are: (1) **FIVs** based on the extreme gradient boosting method can select less crux bands from 400 nm to 1000 nm for the dimension-reduced sub-band dataset. (2) The TPE algorithm can determine the hyperparameters of the RBF-SVM model in the sub-band spectral dataset to perform better than vanilla SVM. (3) The combination of TPE-RBF-SVM and the sub-band dataset selected by **FIVs** can

significantly improve accuracy and F-score metrics compared with two vanilla models and other machine learning algorithms.

In summary, these results show the potential of TPE-RBF-SVM combined with the FIV sub-band dataset for soybean multi-categories recognition. The method in this paper provides a new idea and reference for hyperspectral non-destructive testing for soybean categories.

**Author Contributions:** Conceptualization, Q.Z. and J.F.; Methodology, Q.Z.; Software, Q.Z. and Y.H.; Validation, Q.Z. and Q.Z.; Writing—original draft, Q.Z.; Writing—review and editing, Z.Z., Y.H. and J.F. All authors will be informed about each step of manuscript processing, including submission, revision, revision reminder, etc., via emails from our system or assigned Assistant Editor. All authors have read and agreed to the published version of the manuscript.

**Funding:** This research was funded by the National Key Research and Development Program of China, grant number 2016YFD0300610.

**Institutional Review Board Statement:** Not applicable.

**Informed Consent Statement:** Not applicable.

**Data Availability Statement:** The dataset and fitted models can be downloaded at GitHub at: https://github.com/gniqeh/agriculture-1835236 (accessed on 11 September 2022).

**Conflicts of Interest:** The authors declare no conflict of interest.

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
