# Peer review of "TPE-RBF-SVM Model for Soybean Categories Recognition in Selected Hyperspectral Bands Based on Extreme Gradient Boosting Feature Importance Values"

_agriculture, doi:10.3390/agriculture12091452_

Round 1

Reviewer 1 Report

Dear authors,

The article is very interesting and contains a good piece of work. I have only a few concerns about the paper that could be found in the atachment. I considered that it is minor review.

Author Response

The content in the attached pdf is the same as below.

Point 1: It is the first time that we have seen this acronym. Please provide the holy name for it.

Position: (Page1, Line16)

Response 1: Thank you for this kind note! We will add the full name here. And we have checked all the manuscript to avoid similar mistake.

Point 2: This is not an objective but a methodology.

Position: (Page2, Line93-97)

Response 2: Thank you for pointing out this problem in manuscript. We have a more direct expression there to declare our research objective and adjust some part in Materials&Methods section now.

Point 3&4: This section belongs to Materials and Methods.

Position: (Page3, Line98,99 & Page5, Line171)

Response 3&4: We gratefully appreciate for your valuable suggestion about the whole frame of our manuscript. We have made major adjustments to the original manuscript, mainly the merging of sections and the revising of the order. Now the new article structure is as follows:

1. Introduction
2. Materials and Methods
    2.1. Soybean material and hyperspectral dataset
    2.2. Support Vector Machine Model and Optimization
        2.2.1 Support Vector Machine with Gaussian Radial Basis Kernel
        2.2.2 Optimization of SVM with Tree-structed Parzen Estimator
    2.3. Crux spectrum feature selection based on extreme gradient boosting
    2.4. Feature selection and optimized RBF-SVM modelling
        2.4.1. Feature selection based on feature importance
        2.4.2. Modelling and optimizing RBF-SVM with TPE in sub-dataset
    2.5. Baseline models and evaluation metrics design
        2.5.1 Vanilla and control group models
        2.5.2 Evaluation Metrics and analysis environment
3. Results
    3.1. Feature selection based on FIVs
    3.2. Optimization of SVM by Tree-structed Parzen Estimator
    3.3. Comparison with vanilla model and other algorithms
4. Discussion
5. Conclusion
References

Point 5: The figure must appear after it has been called in the text. It is valid for all figures and tables.

Position: (Page7, Line257)

Response 5: Thank you for this tip! All the figures and tables were checked and we have put them into the right position now.

Point 6: remove dot

Position: (Page7, Line257)

Response 6: Thank you for this detailed note! The typo here has been corrected now. And all similar typos in the manuscript were corrected as well.

Point 7&8: It has not been called in the text… Figure 3 must be here

Position: (Page7, Line264 & Page8, Line268)

Response 7&8: Thanks for your reminding! We have revised it now.

Point 9: In the discussion, you must discuss your results with the literature results. So you will give more foundation to your paper. You can not only discuss your own opinion, but you must consider others' opinions.

Position: (Page14, Line460)

Response 9: Special thanks for your advice! We have re-written the Discuss section now.

(1) We changed discussion path from parallel to progressive arguments and delete the sequence number. According to your suggestion, now the discussion section contains three parts, the necessity of FIVs method and SVM model, the requirement of algorithms combine, and future application scenarios needs

(2) We have adopted a clearer writing way in each discussion paragraph now according to the logical path of:(a) surface phenomenon; (b)inner explanation; (c)contrast with others' work; (d)universality and particularity. As well with special attention as your suggestion, we have combined the references with the research results content now.

Point 10: With your results, please answer if you were able to differ each cathegory of soil been seed. During the results you must present the difereces among each seed cathegory based in your results.

Position: (Page15, Line499*)

Response 10: Thank you for making this valuable suggestion! We have re-written this part according to your suggestion.

(1) Now we summarize clearly in this section that we have accomplished the task of classifying soybeans categories as your comment.

(2) We have made more summary words about the expression of the results. For the specific numerical conclusions such as the values of C and γ, we reserved these into the RESULT section now.

We really appreciate your working for our manuscript,

especially for suggestions of paper frame, Discussion section and Conclusion section!

Reviewer 2 Report

This paper presents a soybean category recognition method using hyperspectral imaging and machine learning. A hyperspectral soybean dataset is collected and created with 2299 soybean seeds and 4 categories. In addition, the authors proposed a method called TPE-RBF-SVM to classify these data. Specifically, extreme gradient boosting is applied to select 10 bands from the 203 bands generated by the hyperspectral camera. Then the selected features are sent to a RBF-SVM model for classification. Then, a Tree-structured Parzen Estimator is employed to optimize the model. Experiment results show the proposed method outperforms some traditional methods. The paper is generally well organized. Concerns of this paper are:

1. All components used in the method: TPE, RBF, SVM and EGB are well established methods in machine learning. So the contribution and novelty is limited.

2. Figure 8 shows MLP2 achieves second best result. This means a shallow neural network performs well on this dataset. Thus, deeper neural network or convolutional network has potential to further improve the results. They should be compared with the proposed RBF-SVM method.

3. There are some deep learning based methods in soybean recognition such as: Li, Hao, et al. "Identification of soybean varieties based on hyperspectral imaging technology and one‐dimensional convolutional neural network." Journal of Food Process Engineering 44.8 (2021): e13767. and Zhu, Susu, et al. "Identification of soybean varieties using hyperspectral imaging coupled with convolutional neural network." Sensors 19.19 (2019): 4065.

4. Will the dataset be open sourced to the research community?

5. English of this paper can be improved.

Author Response

The content in the attached pdf is the same as below.

Point 1: All components used in the method: TPE, RBF, SVM and EGB are well established methods in machine learning. So the contribution and novelty is limited.

Response 1:

Thank you for this very insightful and crux comment.

It is true that we combined mature algorithms in a certain way. But we have a certain application scenario, agricultural detection device designed for soybean. And in the results of other six models (now eight models), only this combine can satisfy our need.

(1) TPE is a novel but practical search algorithm, which is mostly applied in large-scale neural networks. And it is less used for machine learning as an SMBO,or Bayesian optimization, in hyperspectral research where heuristic algorithm is still the mainstream of optimization.

(2) RBF-SVM is a powerful and robust ML method, we hope to balance computing power need and actual performance for certain application field. The TPE combining RBF-SVM could be an auto-machine method but now most core learner of AutoML is Boosting method like H2O, AWS ML stack, etc.

(3) The Extreme Gradient Boosting is usually as the core classifier or regression output, and here we will apply it to feature bands selection for its high interpretability different from traditional methods.

Point 2: Figure 8 shows MLP2 achieves second best result. This means a shallow neural network performs well on this dataset. Thus, deeper neural network or convolutional network has potential to further improve the results. They should be compared with the proposed RBF-SVM method.

Response 2:

Thank you for making this valuable suggestion.

Now the models of deeper multi-Perceptron and convolutional network have been added into the manuscript as the control-group algorithms and the results have been updated. Furthermore, we have an simple analysis for the neural network method in the Section (Result).

(1) adding two deeper NN method as control group model;

(2) updating the results and its explaining.

Point 3: There are some deep learning based methods in soybean recognition such as: Li, Hao, et al. "Identification of soybean varieties based on hyperspectral imaging technology and one‐dimensional convolutional neural network." Journal of Food Process Engineering 44.8 (2021): e13767. and Zhu, Susu, et al. "Identification of soybean varieties using hyperspectral imaging coupled with convolutional neural network." Sensors 19.19 (2019): 4065.

Response 3:

Special thanks for your advice and recommended papers! It is very useful for our team and research!

(1)A careful read and detailed study of the mentioned papers have finished by our team([H.J.]: https://doi.org/10.1111/jfpe.13767 and [L.F.]:https://doi.org/10.3390/s19194065 ). NIR-bands spectrum has such much common with the UV-Vis-NIR in our dataset. And the methods in two papers about the CNN design and configuration help us improve the control-group models as Point2 cited.

(2) We think there is very high reference value in mentioned two paper, so it is necessary to add them into the references of our manuscript.

Point 4: Will the dataset be open sourced to the research community?

Response 4:

Of course! Thanks for your comment about our data transparency.

We thought it would stimulate scientific research in related agricultrue academic field to make the materials in this paper available. Thus, all the data, including dataset, raw pictures, analysis codes, fitted models etc. would be open-source freely under the MIT License after the manuscript finished. That is also why we fixed the random seed as we can to to stress the reproducibility in all processing of analysis. Thank you again!

Point 5: English of this paper can be improved.

Response 5:

Thank you for your sincere comment!

We are apolized for our acdemic English writing ability. The main writing members of this manuscript is not working in an English-speaking environment.

We have proofread the manuscript by a native English speaking colleague. The grammar, spelling, punctuation and phrasing of this manuscript would be checked again. We will consider the Language Editing Services of MDPI if necessary.

We really appreciate your working for our manuscript!

Round 2

Reviewer 2 Report

The authors have addressed all my comments.